# A Practical Framework for Academics to Implement Public Engagement Interventions and Measure Their Impact

**DOI:** 10.3390/ijerph192013357

**Published:** 2022-10-16

**Authors:** Isolde Martina Busch, Silvia Savazzi, Giuseppe Bertini, Paola Cesari, Olivia Guaraldo, Michela Nosè, Corrado Barbui, Michela Rimondini

**Affiliations:** 1Section of Clinical Psychology, Department of Neuroscience, Biomedicine and Movement Sciences, University of Verona, 37134 Verona, Italy; 2Perception and Awareness Laboratory, Department of Neuroscience, Biomedicine and Movement Sciences, University of Verona, 37134 Verona, Italy; 3Section of Anatomy and Histology, Department of Neurosciences, Biomedicine, and Movement Science, University of Verona, 37134 Verona, Italy; 4Section of Movement Sciences, Department of Neuroscience, Biomedicine and Movement Sciences, University of Verona, 37134 Verona, Italy; 5Department of Human Sciences, University of Verona, 37129 Verona, Italy; 6Section of Psychiatry, WHO Collaborating Centre for Research and Training in Mental Health and Service Evaluation, Department of Neuroscience, Biomedicine and Movement Sciences, University of Verona, 37134 Verona, Italy

**Keywords:** public engagement, third mission, knowledge transfer, co-creation, research impact, implementation science, framework, critical thinking, scientific literacy, public understanding of science

## Abstract

Academic institutions have shown an increased interest in the so-called third mission to offer an impactful contribution to society. Indeed, public engagement programs ensure knowledge transfer and help to inspire positive public discourse. We aimed to propose a comprehensive framework for academic institutions planning to implement a public engagement intervention and to suggest potential indicators to measure its impact. To inform the framework development, we searched the literature on public engagement, the third mission, and design theory in electronic databases and additional sources (e.g., academic recommendations) and partnered with a communication agency offering non-academic advice. In line with this framework, we designed a public engagement intervention to foster scientific literacy in Italian youth, actively involving them in the development of the intervention. Our framework is composed of four phases (planning/design, implementation, immediate impact assessment, and medium- and long-term assessment). Impact indicators were subdivided into outcome variables that were immediately describable (e.g., changed understanding and awareness of the target population) and measurable only in the medium or long run (e.g., adoption of the intervention by other institutions). The framework is expected to maximize the impact of public engagement interventions and ultimately lead to better reciprocal listening and mutual understanding between academia and the public.

## 1. Introduction

Aside from teaching and research activities, academic institutions have recently shown an increased interest in the so-called third mission—a term used especially in the European academic context—to offer an impactful contribution to society [1,2,3,4] by transmitting knowledge and technological achievements and innovations to the industrial sector and the public [2]. Molas-Gallart and colleagues [5] (pp. iii–iv) provided a rather broad definition of the concept of a third mission, stating that it is “concerned with the generation, use, application and exploitation of knowledge and other university capabilities outside academic environments. In other words, [the third mission] is about the interaction between universities and the rest of society”.

The more than 200 land grant colleges and universities in the United States follow a very similar concept, albeit named differently [6,7]. Fulfilling their so-called “extension mission”, these land grant institutions do not only teach and conduct research but “extend” their resources, disseminating results to the public, offering research-based programs and courses on various relevant topics, and providing practical support with the aim of improving the lives of individual persons, families, and communities within their state [6,7].

Public engagement and involvement programs, which are an important part of third mission activities, ensure knowledge transfer, help to inspire positive public discourse, and encourage interaction between researchers and citizens. These interventions can help to instill curiosity about science in citizens and foster scientific literacy, critical thinking, and scientific confidence. On the other hand, the public can help to drive new research questions by sharing precious insights and experiences (see Figure 1).

Concrete examples of public engagement activities include online and in-person lectures for the general public [8], academic podcasting [9], digital engagement activities [10], science festivals (e.g., the European Researchers’ Night [11]), and book exhibitions [12], as well as activities for children and adolescents, such as STEM programs for girls or marginalized youth [13,14], hands-on art-making activities [15], kids’ universities [16], and family science workshops [17].

Similarly, patient engagement has recently become a key principle of patient-centered healthcare, fostering patient independence, and likely resulting in better health outcomes (e.g., patient compliance) [18,19,20,21]. Actively listening to patient experiences and insights can also help to drive new research directions, encourage innovation, and improve patient safety [22,23]. Further, building community–academic partnerships, as is done for instance at the Johns Hopkins Hospital in the United States, can help to establish health equity in marginalized communities [24,25].

To be effective and tailored to the respective target population, third mission activities should follow systematic approaches. Although several general methods for strategic project planning and management exist, up to now, too little attention has been paid to the design of comprehensive techniques for the development of public engagement interventions. Therefore, we aimed to propose a detailed, structured framework for academic institutions planning to implement public engagement interventions; to suggest potential indicators to measure its immediate-, medium-, and long-term impact; and to outline methods for dissemination.

## 2. Development of the Framework

To develop the framework, we examined current theories of design thinking, service design [26,27,28,29,30], and project management [31,32,33,34,35] and conducted a comprehensive literature search on the concept of universities’ third mission and public engagement interventions, including database scans (i.e., PubMed, Web of Science Core Collection) and grey literature searches (e.g., websites of universities and non-profit organizations [8,9,10,11,12,13,15,16,36,37,38,39,40,41,42,43,44,45,46,47,48,49]). The search terms included “Third mission”, “public engagement”, “co-creation”, and knowledge transfer”. Further, we searched national and international sources discussing impact indicators (e.g., Third Mission Guidelines of the National Agency for the Evaluation of Universities and Research Institutes (Agenzia Nazionale di Valutazione del Sistema Universitario e della Ricerca—ANVUR [50]), REF Impact Case Studies Databank [51]).

We then partnered with a Verona-based communication agency (Forest Hand), which offered non-academic advice and valuable insight into the field of communication studies and science communication during several meetings. In particular, the agency provided practical recommendations for the development of the framework and shared their experience in the application of the canvas model in different settings. A set of meetings was organized in which our research team was invited to complete the canvas model under the supervision of the communication agency. In these meetings, questions, critical issues regarding the canvas model, and the timeline and structure of the framework were discussed and resolved.

Finally, we determined the critical variables discussed in the retrieved material, reflected on their relationships, and used the agency’s practical suggestions to construct the framework [22,52,53].

## 3. Framework for Public Engagement Interventions

The framework is subdivided into four phases (i.e., planning and design; implementation; and immediate-, medium-, and long-term impact assessment) with different steps (Figure 2). It focuses on the engagement of stakeholders, the target population, and the wider public. It should be noted that it is only indicative and may be updated according to demand. In the following, the different phases of the framework are described.

### 3.1. Planning and Design Phase

The research group should allocate an appropriate amount of time to this first phase, as many aspects must be discussed and critical decisions made.

Conducting a situational analysis at the very beginning is critical when assessing whether the public engagement intervention is warranted [31]. In most cases, it includes the SWOT analysis, and if the long-term effects of the program shall be examined, PEST analysis is undertaken to ensure the success of the project and its sustainability. These two techniques complement each other and are often used in conjunction [32].

Indeed, a SWOT analysis is a strategic planning method that is used to detect and evaluate certain factors (i.e., internal strengths and weaknesses, external opportunities and threats) linked to an organization or project that might influence current tasks and future activities [33]. In contrast to the SWOT analysis, the PEST analysis looks at a bigger picture, namely, the political, economic, socio-cultural, and technological changes the organization or project might be exposed to. This allows for the detection of opportunities and the assessment of current and future risks and threats [32].

Both SWOT and PEST analyses provide a rich background for the selection of the main topic and the objectives of the intervention. In this early phase, the research group should remain open to exploring different ideas for topics and reviewing the related research. Indeed, if there are several preliminary topics that seem to be equally interesting and choosing one is difficult, it is advisable to continue with the other steps in the *planning and design phase* and circle back to the question later.

To define the objectives of the public engagement intervention, the researchers need to ask themselves what they want to achieve and what kind of difference they want to make. Do they want, for instance, to increase knowledge, expand awareness, receive feedback on their own research, change points of view, or promote critical thinking? It is also recommended to define SMART aims (i.e., specific, measurable, achievable, relevant, and time-based) [54].

Defining the aim(s) is a particularly important step, as it is closely linked to another step in the planning phase, namely, the identification of the impact. Indeed, when discussing the potential aims of the intervention, the research group might consider whether they want to achieve a broad impact (i.e., the activity/intervention reaches a large group of people), a deep impact (i.e., the activity/intervention addresses a small number of people), or they point to both types of impact. In the latter case, for instance, a combination of activities addressed to both small and big groups is possible [55].

When defining the objectives of the intervention, the research group should be aware that the same goal can often be achieved in different manners [34]. Thus, the use of the so-called analysis of alternatives (AOA) is suggested throughout the planning and design phase. This specific technique evaluates the various options that are available for the achievement of a specific objective. It can help to identify the most resource-oriented option that has the highest probability to reach a certain desired impact. Methods such as life-cycle costing and cost-benefit analyses are often applied in the course of the AOA [35].

The identification of the target group (i.e., the audience that will be actively involved in the public engagement intervention and benefit from it), and if necessary, of certain subgroups, depends on the chosen topic and the public engagement aims. The research group should try to be as specific as possible in doing this since a detailed description of the specific target will also help later to identify the related needs and the stakeholders that must be involved.

When defining the target group, the research group should ensure that they are inclusive and accessible, foresee potential barriers to attending and participating for some groups (e.g., physical barriers, financial barriers, concerns among LGBTQ+ people regarding anonymity, learning difficulties), and offer support and encouragement [38,48,56]. Consulting the involved stakeholders throughout the project and gathering feedback from the participants can help the research group to navigate this at times challenging path [56]. Reed [57] also underlined how important it is to be humble when engaging with the stakeholders and the target population to remove barriers and hierarchies and acquire trust.

Indeed, stakeholders (i.e., people or organizations who have an interest in the public engagement activity or who affect or are affected by its outcomes [58]) can and should serve as consultants. The research group might even want to establish a stakeholder advisory panel for this purpose [57]. While stakeholders do not participate in the activity like the target group does, they are largely involved in its making and play an important role in the success of the project. The stakeholders often act as “facilitators”, building links between the research group and the target population and helping to define the needs of the target group. Therefore, stakeholders should be chosen carefully. Correct identification of the target group beforehand can give a useful indication of the potential stakeholders. In some cases, it might be useful to involve not only the stakeholders whose interests are directly linked to the topic of the intervention but also other people, such as investors, collaborators, and volunteers (e.g., funders, charities, professional bodies) who can provide precious resources, such as time, expertise, or funds, as well as original thoughts and an outside point of view [59].

To better understand the problems and actual needs of the target group, which is crucial for the success of the project, the research group should directly address the target group and stakeholders when gathering both qualitative and quantitative information while of course adhering to ethical guidelines and ensuring confidentiality and anonymity [60].

A preliminary investigation into the background knowledge, attitudes, preferences, and experiences of the target audience can help in this endeavor. Further, any doubts that might have emerged during the first steps of the planning and design phase are likely to be resolved with such an investigation. The type of investigation (e.g., (semi)structured interviews, focus groups, surveys, online polls) depends also on whether it is suitable for the target group [48].

The type of event or activity should be informed by the specific characteristics of the target group (e.g., age and social background). Moreover, the results of the preliminary investigation can also be useful for choosing the activity. There are various ways to engage with the participants actively and productively, such as through games, writing, science exhibitions and performances, digital platforms, podcasts, and videos.

Then, the activity needs to be carefully planned, considering logistical aspects (e.g., location, equipment, budget, and internal and external funding) and ethical issues (e.g., safety, anonymity, confidentiality, and maintenance of professional integrity) [60]. Moreover, the content that will be conveyed must be determined. In all of this, the researchers should strive toward designing an intervention that is accessible, enjoyable, and enriching [61]. To ensure that the intervention is attractive, it might be useful to collect feedback from some people that are part of the target audience [62].

Another issue that needs to be thought through clearly is the budget (i.e., expenses and any budget that is already allocated and therefore available). Creating an itemized budget is useful for gaining a clear picture of the implementation cost items of the projects (e.g., reimbursements, acquisition of equipment, production of promotional material, and costs of the venue) [63]. While funding is often difficult to secure, it is worth watching out for options for internal (e.g., funding by the university) and external funding (e.g., funding by external agencies and non-profit organizations).

Finally, the identification of specific indicators that will describe its immediate-, medium- and long-term impact throughout and after the intervention should also already take place in this early phase. It is pivotal that the identified impact indicators, in particular, the medium- and long-term impact indicators, are objective and SMART (i.e., specific, measurable, achievable, relevant, and time-based).

Figure 3 provides an overview of such indicators.

As illustrated, indicators that are immediately describable encompass, for example, the number of main beneficiaries reached (e.g., members of the target population, individuals inside and outside of the academic setting, social media data (e.g., number of shares, likes, mentions, followers, hashtag usage, and URL clicks) [64], the efficacy of the intervention rated by the target population, and the learning outcomes. Regarding the latter, the learning outcomes for the target group include different aspects, namely, enhanced knowledge and understanding, skills (e.g., intellectual skills, social skills, communication skills, and physical skills), attitudes and values (e.g., feelings, increased capacity for tolerance, and increased motivation), enjoyment, inspiration, creativity, behavior, and progression (e.g., reported actions and changes in lifestyles) [65].

While an assessment of the medium- and long-term impact might not be needed for or be applicable to every type of public engagement intervention, it can in many cases be very useful and informative.

Medium-term impact indicators include the dissemination of findings to the public (e.g., number of dissemination events and webinars and number of participants, newspaper articles, and radio shows) and the academic community (e.g., number of publications in national and international peer-reviewed journals, number of citations, and number of downloads of the publications). Finally, there are indicators that are usually only describable in the long run (e.g., the adoption of the public engagement intervention by other universities, educational institutions, public and/or charity organizations, and the influence of new public policies). Obviously, the different categories (i.e., immediate-, medium-, and long-term) cannot always be strictly separated and smooth transitions are possible. For instance, in some cases, the overall economic value of external financing might be determinable only in the medium- or long-term and not immediately after the completion of the intervention.

The Impact of the public engagement intervention can be described not only in terms of time but also in terms of affected areas. As defined by the Higher Education Funding Council for England, an impact is “an effect on, change or benefit to the economy, society, culture, public policy or services, health, the environment or quality of life beyond academia” [57] (p. 28). Figure 4 portrays these areas of positive impacts of public engagement interventions and offers examples of each area. While economic impacts, impacts on health and well-being, impacts on policy, and environmental impacts are rather self-explanatory, cultural impacts and social impacts are more difficult to describe. Although seemingly like cultural impacts (i.e., changes in predominant values, opinions, and behavior patterns benefitting organizations, social groups, or society in general [57]), social impacts regard any changes that improve social injustices and overcome related barriers [66,67].

#### Public Engagement Model Canvas

The main elements of the proposed intervention that are discussed in the planning and design phase (i.e., topic, target population, stakeholders, needs of the target population, objectives of public engagement intervention, type of event, preliminary investigation, budget, and measurement of impacts) can be gradually inserted into the so-called Public Engagement Canvas. Subdivided into several quadrants, the canvas helps in the creative development and realization of public engagement activities by providing a comprehensive and visual overview (Figure 5).

The canvas can be used as helpful guidance and a reference point not only in the initial planning and design phase but also in the following phases. Namely, after a careful rereading of the inserted elements and a check for congruency, the research group can share the canvas, along with a synthesized report, with the involved stakeholders and external bodies, or archive it internally. It can, if needed, always be modified and updated with any new, relevant information. In some cases (e.g., when there are two different target groups), it is preferable to work on two canvasses in parallel.

The canvas is based on service design and design thinking theories, which promote “outside the box” thinking and foster co-creation processes [28,29,30]. Design thinking, which is composed of different phases (i.e., inspiration, ideation, implementation), aims to tackle problems in novel and inventive ways by applying the principles of empathizing with human needs, defining the problem, ideating, prototyping, testing possible solutions [26,27,29], and to offer immediately available and easily applicable tools for problem-solving.

Aside from the Public Engagement Model Canvas, it is also recommended to prepare a Gantt chart to track the progress of the project. Finally, a dissemination plan should already be drafted in this first phase that identifies dissemination targets (e.g., target group(s) of the intervention, involved stakeholders, and demographic subgroups) and channels (e.g., mailing lists, a dedicated website, and social media). The plan will then be refined and deepened and/or expanded in the immediate impact assessment phase.

### 3.2. Implementation Phase

From this phase on, the research group should repeatedly use the Public Engagement Model Canvas as a reference point. If any uncertainties regarding the intervention remain, an additional preliminary investigation, like the one conducted in the planning and design phase, can be performed.

Before launching the event, the research group must thoroughly organize it following the plan established in the planning and design phase. Depending on the activity, materials and tools might have to be prepared, equipment sorted, and venues booked.

The involved stakeholders should be regularly briefed and updated about the work progress given to the involved stakeholders as well as to the wider public through the previously defined dissemination channels [67]. Sharing your work with the public at large early on can have several advantages, as it can, for instance, enhance awareness of the project, and thus, the chance of additional funding, and provide learning opportunities and inspiration for other academics [67].

### 3.3. Immediate Impact Assessment Phase

To collect evidence of the immediate impact, including the learning outcomes and satisfaction measures, as described above, the research group should design and conduct a post-intervention evaluation with the target population using, for instance, surveys and/or semi-structured interviews. Specific questions addressed to the involved stakeholders regarding, for instance, their experiences, satisfaction with the public engagement intervention, and potential barriers and challenges, can also yield interesting additional results and new, diverse perspectives.

Additionally, certain impact indicators that are not directly collected from the target population and its stakeholders might need to be assessed. The subsequent synthesis of all immediate impact indicators serves different purposes, namely, for internal documentation and dissemination activities, and as a basis for a scientific publication. Then, following the refined dissemination plan, dissemination activities and events need to be organized. Reaching a broad audience through such events as well as through social media is likely to result in a greater medium- and long-term impact of the intervention [37]. Dissemination events are also great opportunities to make those involved in the public engagement intervention feel valued and show that their efforts have been appreciated [39]. The research group should also ensure that the achievements of the target population are sufficiently acknowledged during these events.

### 3.4. Medium- and Long-Term Impact Assessment Phase

Between the third and this fourth, last phase, a certain time interval will pass since the respective impact indicators (e.g., recommendation of the intervention by public and/or private agencies and influence on public policy) are, as outlined above, only measurable in the medium and long run. Additional to the post-intervention evaluation during the preceding phase, a final meeting with stakeholders and members of the target population can prove valuable to discuss the significance and reach of the generated impact [68] and identify areas for improvement. Moreover, it is a good opportunity to appreciate the value of the work from all involved partners [54].

In the last step, the research group might decide to write a final report that is addressed to stakeholders, the target group, university bodies, and/or the public that discusses the achieved medium- and long-term impacts and share the results of the report on traditional and/or digital channels. These insights can inspire others to implement this proposed public engagement intervention or can help colleagues who are planning other types of public engagement activities to maximize their impact [54,55]. The research group should avoid breaking off relations with the target population and the stakeholders at the end of the project. Staying connected not only prevents the involved persons from feeling suddenly let down by the project organizers but also encourages ongoing knowledge exchange [57].

### 3.5. Critical Issues and Limitations

During project development, researchers should also always keep in mind several critical issues that they might encounter along the way. Indeed, a survey by the University of Reading, UK, detected several main challenges and barriers to public engagement activities [40]. The authors listed as potential challenges, inter alia, the availability and continuity of funding, public engagement (e.g., difficulty of maintaining long-term public engagement and identification of target audiences), allocation of time to the project, lack of recognition of public engagement work by the academic community and government authorities, poor communication between stakeholders, lack of best practices and examples of similar projects, and management of expectations of involved researchers and stakeholders [40].

We are aware that the proposed framework and the Public Engagement Model Canvas may have certain weaknesses that researchers should be aware of in the project design. One concern regarding the framework might be that it is only indicative and overall too general. However, given the variety of public engagement activities, we wanted to ensure that it is flexibly applicable to different types of events. Regarding the Public Engagement Model Canvas, it might appear too static since it does not reflect changes in strategy and the evolvement of the intervention [69]. To address this issue, the research group should modify the canvas according to demand, which is a procedural step that certainly requires an additional time investment. Finally, as highlighted in the literature [57,70], authors run the risk of overestimating the impact and reach of their intervention and of not clearly articulating the actual benefits and identifying the beneficiaries.

## 4. Real-World Example: Debunker—A Public Engagement Intervention to Foster Critical Thinking and Scientific Literacy in Italian Youth

In the following, we present Debunker, a newly developed public engagement intervention for young people that serves as a real-world example, and briefly describe the completed planning and design phase and outline the upcoming phases of implementation, impact assessment, and dissemination.

### 4.1. Planning and Design Phase

After brainstorming ideas for creative and valuable public engagement interventions and narrowing them down to a few options, our research team agreed on the topic, aim, and type of the intervention; defined the target population and the stakeholders; created an itemized budget; and identified several impact indicators. Along with these considerations, we performed SWOT and PEST analyses, analyzing the various smaller and bigger factors that could influence our public engagement intervention. Subsequently, we started to carefully review the relevant literature and again consulted our partner, the Verona-based communication agency Forest Hand, which offered an outside perspective and gave us critical feedback on our proposal.

As a next step, by administering an ad hoc survey via appropriate communication channels (e.g., WhatsApp, mailing lists, and social media), we will identify the science- and medicine-related topics (e.g., verification of scientific soundness of newspaper articles, vaccine hesitancy, climate change, and body image on social media) that are of particular interest for our target population, which was recruited from high schools and sports clubs. Before proceeding to outline the next operational steps, the research underpinning Debunker and our main aim are briefly described.

### 4.2. Underpinning Research and Main Aim of Debunker

The COVID-19 pandemic and its accompanying “infodemic” has shed a bright light on matters of pressing concern of our time [71,72,73,74,75], that is, a significant and increasing difficulty by large parts of the population to correctly distinguish facts, misinformation, and disinformation, which, for some people, can culminate in openly embracing science denialism [76,77,78,79,80]. This phenomenon is amplified by a gradual loss of trust in official information sources in favor of a “tailor-made” selection of informal sources of information that tend to only confirm one’s own beliefs and suspicions [75,81,82,83]. Academic institutions play a crucial role in addressing this issue. Indeed, universities following the third mission can help to ensure knowledge transfer and to encourage dialogue between academia and the public [2,4]. Considering the constant exposure of young people to digital media with its rapidly changing online news cycles and potentially harmful social media algorithms, as recently highlighted by the Wall Street Journal Investigation “The Facebook Files” [83,84,85,86,87], it is particularly important to provide young people with appropriate instruments to better face the complexity of digital content and to foster, by appealing to their intrinsic curiosity for the world [88], an open-minded and critical approach to science and knowledge. Therefore, we aimed to design and implement a public engagement intervention to foster critical thinking and scientific literacy in Italian youth, actively engaging our target group in the development of the intervention.

### 4.3. Public Engagement Model Canvas

Following the proposed framework, we prepared a Public Engagement Model Canvas to easily visualize the main elements of Debunker (see Figure 6). However, it represents only a first, preliminary version since, as stated above, we have not yet assessed the needs of the target group.

After designing the canvas and critically revising its content, we shared it, together with a short report, with university bodies and all stakeholders involved in Debunker (i.e., university researchers, the communication agency “Forest Hand”, and contact persons from schools and sports clubs). After the needs assessment of the target group, we will update the canvas accordingly. Further, we prepared the first drafts of the Gantt chart and the dissemination plan that might be revised at a later stage of the project.

#### 4.3.1. Implementation Phase

Based on the outcomes of the preliminary investigation, and thus, being closely tailored to young people’s priorities and needs, we will create four videos that will be shown to different age groups (ages 11–14, 15–17, 18–21).

During this phase, we will brief the stakeholders on a regular basis and start sharing updates of our work on traditional (e.g., press releases) and/or digital channels (e.g., dedicated website of our Debunker project, Twitter, Instagram, Facebook, and LinkedIn).

#### 4.3.2. Immediate Impact Assessment Phase

In this phase, we will employ a post-survey to assess the intervention impact and its different dimensions.

To measure the nature and extent of the impact of the intervention, several impact indicators identified in the planning and design phase (e.g., participants’ satisfaction, acquired knowledge, improved awareness, and the degree to which participants’ needs were considered) will be assessed by the post-survey. The post-survey data will then be synthesized by the research team for dissemination activities and events.

To ensure a broad reach of the intervention and the related findings, we will refine and expand the dissemination plan drafted in the planning and design phase while also following the previously identified impact indicators. Dissemination activities will target young people, stakeholders, the public, and the scientific and healthcare community. A dissemination event addressing the public and, in particular, adolescents, young adults, school representatives, teachers, coaches, and journalists, will be organized to present Debunker and to discuss the related findings and implications. Brochures providing an overview of the intervention will be distributed during this event. Further, all members of the target population will receive a certificate as a symbol of their active participation, commitment, and achievements in scientific literacy.

Further, we aim to publish articles in local newspapers describing our intervention and its outcomes. Academics and healthcare providers will be reached through scientific publications in national and international peer-reviewed journals, as well as oral presentations and posters at national and international scientific conferences. A dedicated website and social media posts (Twitter, Facebook, Instagram, etc.) will address both the scientific community and the public.

#### 4.3.3. Medium- and Long-Term Impact Assessment Phase

We also expect positive intervention effects in the medium-to-long term that cannot be measured immediately after the intervention but only in successive phases. Examples of such impact indicators include public dissemination of the findings, citations in the scientific literature, likes and shares on social media, adoption of the intervention by other universities and educational institutions, and the launch of new projects and interventions inspired by Debunker. Finally, a future paper will provide a detailed description of our public engagement intervention and report the results of the preliminary investigation and the post-survey, as well as the assessment of the immediate-, medium-, and long-term impacts.

### 4.4. Expected Challenges during the Implementation of Debunker

As mentioned above, one of the challenges in the field of public engagement is the appropriate allocation of time to the project. Thus, we will have to be mindful of the fact that the preparation of videos requires a lot of time and focus. Our team will also have to be aware of the potential risks that might arise when using the Public Engagement Model Canvas, as mentioned above. For instance, because of the canvas’s relatively static structure [69], it will be important to frequently check whether it is still up to date and, if not, modify it and brief the involved stakeholders accordingly. Moreover, as videos may promote less interaction with the audience than other types of public engagement interventions [89], we will have to make sure to regularly collect feedback from stakeholders and participants and organize in-person events to connect and share insights. This will also help with maintaining the long-term engagement of our target group, which was one of the challenges pointed out by the University of Reading [40].

## 5. Conclusions

We believe that the proposed framework and canvas serve as a beneficial reference for academic centers being committed to proactively engaging with society. Indeed, research groups aiming to implement different kinds of public engagement interventions are guided through all different process stages, from planning/design to the actual implementation to the assessment of the impact and dissemination of the outcomes.

The presented systematic method, which stands out due to its comprehensiveness, is expected to maximize the significance and reach of the impact in different areas and ultimately lead to improved reciprocal listening and mutual understanding between academia and the wider public.

## Figures and Tables

**Figure 1 ijerph-19-13357-f001:**
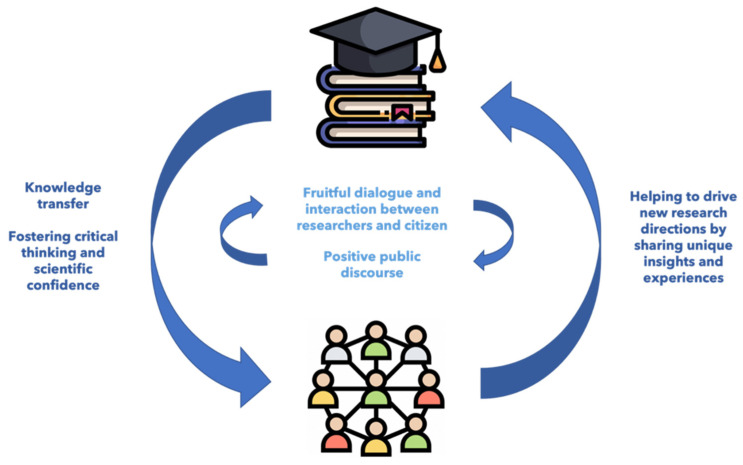
Public engagement as part of third mission activities.

**Figure 2 ijerph-19-13357-f002:**
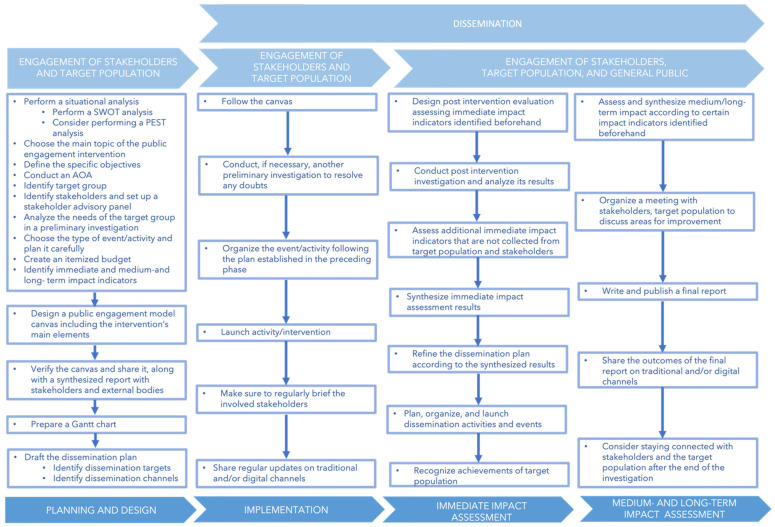
Framework for public engagement interventions. Abbreviations: AOA, analysis of alternatives; PEST, political, economic, socio-cultural, and technological changes; SWOT, strengths, weaknesses, opportunities, and threats.

**Figure 3 ijerph-19-13357-f003:**
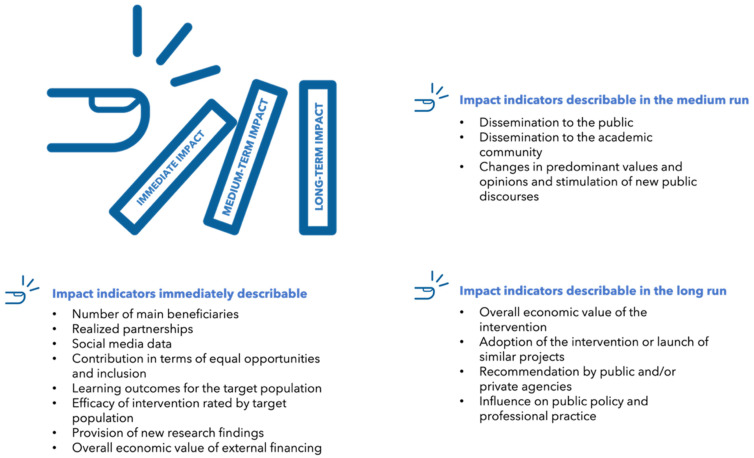
Immediate-, medium-, and long-term impact indicators.

**Figure 4 ijerph-19-13357-f004:**
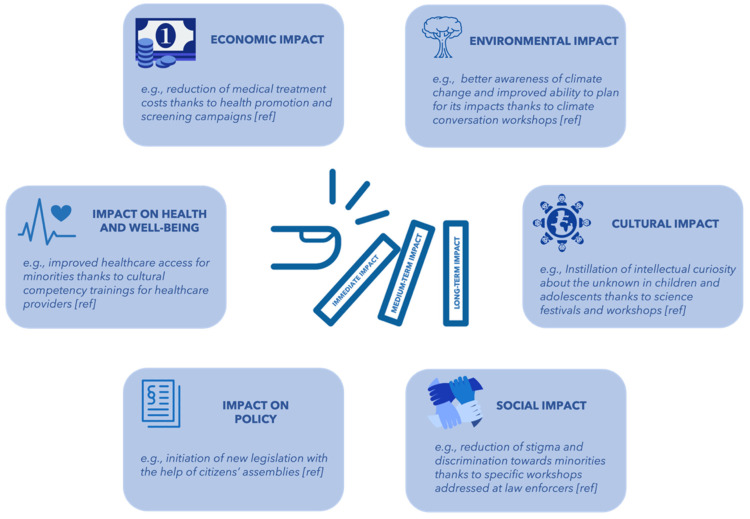
Areas potentially impacted by public engagement interventions. Note: the money icon was made by Paomedia (https://www.iconfinder.com/icons/299107/money_icon), the environment icon was made by Maxicons (https://www.iconfinder.com/icons/5360441/environment_forest_natural_nature_tree_wood_icon), and the colors were modified by the authors of this paper. Accessed on 30 May 2022.

**Figure 5 ijerph-19-13357-f005:**
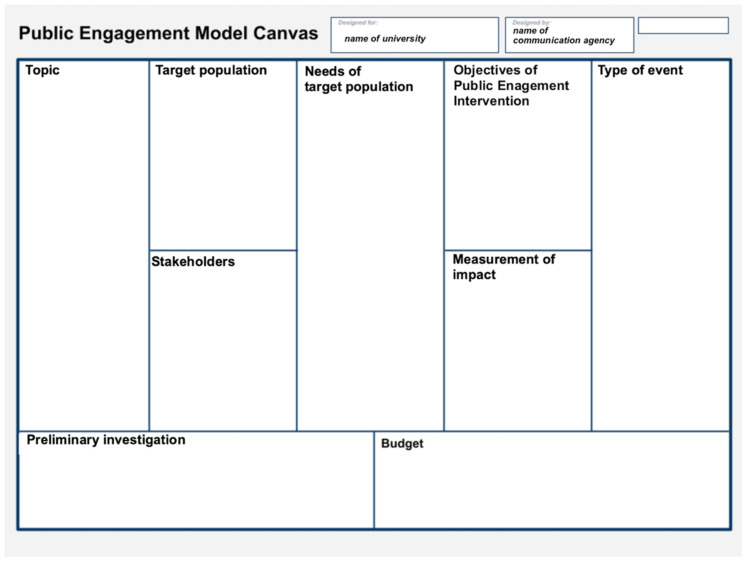
Public Engagement Model Canvas.

**Figure 6 ijerph-19-13357-f006:**
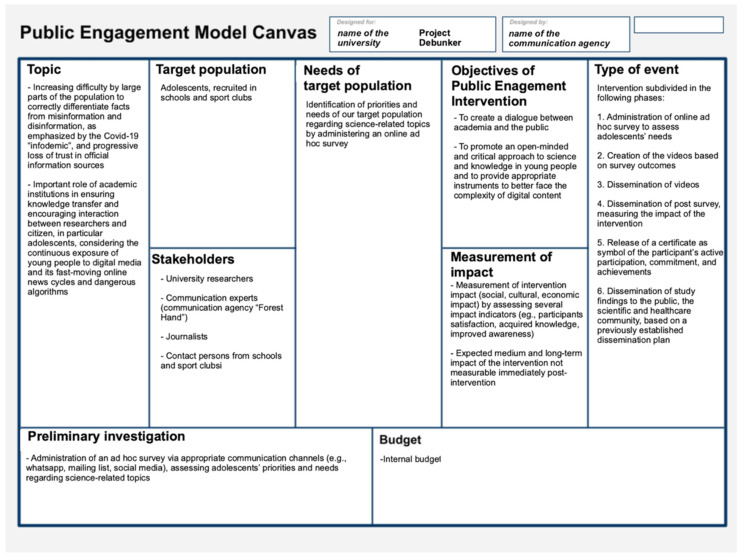
Public Engagement Model Canvas adapted to Debunker.

## Data Availability

Not applicable.

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
