# Peer review of "A Practical Framework for Academics to Implement Public Engagement Interventions and Measure Their Impact"

_ijerph, 2022, doi:10.3390/ijerph192013357_

Round 1

Reviewer 1 Report

I appreciate being given the chance to evaluate this work. Although it has scientific merit, further explanation is still needed to improve understanding.

Major concern:

The manuscript's title is excessively long. The text does not adequately clarify vital key words like "transferring scientific evidence to the public."

The conceptual framework for an intervention program resembles the current framework in theory. To justify the study's title, the knowledge transfer component was not prominently highlighted or given top priority.

How much does the conceptual framework that was created differ from the one that is already in place? What new value was brought to the framework?

As usual, a situational analysis or need analysis should come first when we want to implement a community intervention program. It needs to be given top priority. Most of the time, it comprises SWOT analysis, and if the program's long-term effects are being examined, PEST analysis should also be considered to assure the project's success and sustainability. The need analysis is listed in the middle of the list in Figure 2, where the authors attempt to prioritize the main topic and identify the objective first.

Minor concern:

1.       Figure 3 – the medium- and long-term impact indicators need to be objective and SMART.

2.       Figure 2 – it is suggested to include one more stage – choosing the alternative project method to measure the impact of the intervention. The selected tools must undergo a thorough validation process.

3.       In other words, [it] is about the interaction between universities and the rest of society.” [p.iii-iv] – can you explain the citation in this sentence (line 52 – 53)

Reviewer 2 Report

The manuscript submitted by Busch et al., is presenting an interesting conceptual approach on a much debated and discussed topic regarding the mission of academia towards the service of society in addition to the teaching and research missions already fairly well established and understood. 

The approach is interesting and well presented. The reviewer would like to point out two major points for improvement.

1. While the authors do occasionally refer to US Academia, the manuscript would benefit significantly if it discussed the concept of Extension that Land Grant Universities have in the the US. There is a long standing tradition and perception of public service as well as dissemination of results to the public and practical support by Universities through their Extension mission.

2. Proofreading for correcting typos and improving grammar and syntax/flow of the narrative of the manuscript.

Nice work overall!

Round 2

Reviewer 1 Report

I am satisfied with the quantity of correction made, and I recommend accept.

Authors are to be congratulated to their excellent research.